# LncRNA HULC and miR-122 Expression Pattern in HCC-Related HCV Egyptian Patients

**DOI:** 10.3390/genes13091669

**Published:** 2022-09-18

**Authors:** Dalia A. Gaber, Olfat Shaker, Alaa Tarek Younis, Mohamed El-Kassas

**Affiliations:** 1Medical Biochemistry & Molecular Biology Department, Faculty of Medicine, Helwan University, Cairo 11795, Egypt; 2College of Medicine, Gulf Medical University, Ajman 4184, United Arab Emirates; 3Department of Medical Biochemistry & Molecular Biology, Faculty of Medicine, Cairo University, Cairo 11956, Egypt; 4Endemic Medicine Department, Faculty of Medicine, Helwan University, Cairo 11795, Egypt

**Keywords:** miR-122, HULC, HCC, HCV, molecular biomarkers

## Abstract

Hepatocellular carcinoma (HCC) is a highly prevalent malignancy. It is a common type of cancer in Egypt due to chronic virus C infection (HCV). Currently, the frequently used lab test is serum α-fetoprotein. However, its diagnostic value is challenging due to its low sensitivity and specificity. Genetic biomarkers have recently provided new insights for cancer diagnostics. Herein, we quantified Lnc HULC and miR-122 gene expression to test their potential in diagnosis. Both biomarkers were tested in the sera of 60 HCC patients and 60 with chronic HCV using real-time RT-PCR. miR-122 was highly expressed in HCV patients with a significant difference from the HCC group (*p* = 0.004), which points towards its role in prognosis value as a predictor of HCC in patients with chronic HCV. HULC was more highly expressed in HCC patients than in the HCV group (*p* = 0.018), indicating its potential use in screening and the early diagnosis of HCC. The receiver operating characteristic (ROC) curve analysis showed their reliable sensitivity and specificity. Our results reveal that miR-122 can act as a prognostic tool for patients with chronic HCV. Furthermore, it is an early predictor of HCC. LncRNA HULC can be used as an early diagnostic tool for HCC.

## 1. Introduction

Primary liver cancer is considered the sixth most commonly diagnosed cancer and the third leading cause of cancer death worldwide according to the IARC (International Agency for Research on Cancer) in 2020, with higher incidence rates among men. The highest prevalence rates are seen in transitioning countries, including Egypt [1].

Hepatocellular carcinomas represent almost 70–85% of all primary liver cancers, which are the most common histological form [2]. Therapeutic options for HCC patients are limited and the disease is often diagnosed at an advanced stage. Hepatocellular carcinoma can only be eliminated in its early stages by surgery; hence, the prognosis is often bad [3]. Recently, there has been robust progress in molecular targeted therapies. These advances have shifted the treatment modalities for advanced HCC. Since then, it has become of utmost importance to find biomarkers that can predict treatment response, hence providing guidance towards the strategy of systemic therapy [4].

Genetic and epigenetic alterations are considered the main drives for hepatocyte malignant transformation. Despite the huge continuous efforts and advances in therapeutic strategies, HCC survival rates remain low and poorly understood [5].

Non-coding transcripts of less than 200 bases are called small non-coding RNAs. This category includes the following: tRNA, rRNA, miRNA, snoRNA, and piwi-interacting RNA (pi-RNA). Moreover, RNA molecules of more than 200 bases in length are scientifically known as long non-coding RNAs. lncRNAs, miRNAs (21–24 bases), and piRNAs (26–31 bases) function as regulators of gene expression on both the transcriptional and post-transcriptional level. They also play a role in DNA epigenetic modification.

Both lncRNAs and miRNAs are non-coding. MiRNAs are about 22 nucleotides long, whereas lncRNAs are 8–10 times longer. Although the exact functions of lncRNAs are not yet scientifically well defined, some functions are known for both miRNAs and lncRNAs as they act as regulators to control some biological processes through the post-transcriptional repression of protein-coding genes. LncRNAs can also act as miRNA sponges, thus decreasing the miRNA regulatory effect of mRNA. Scientists have identified nearly 2000 different miRNAs and about 50,000 lncRNAs [6].

Thus, the identification of novel, reliable, and non-invasive biomarkers for HCC allows a detailed understanding of the molecular mechanisms underlying hepatic carcinogenesis, as well as providing a tool for early diagnosis, which improves patients’ outcomes. Herein, we conducted this study to examine the possibility of relying on new early biomarkers in both hepatocellular cancer patients and hepatitis C patients to be used for screening and early diagnosis.

## 2. Patients and Methods

### 2.1. Study Design

This prospective, cross-sectional study was conducted between January 2021 and February 2022 at the department of Medical Biochemistry and Molecular Biology, Faculty of Medicine, Cairo University, Egypt. This study was approved by the Research Ethics Committee of the Faculty of Medicine, Helwan University, Egypt. Informed consent was obtained from all participants.

### 2.2. Study Groups

Sixty patients who were diagnosed with hepatocellular cancer (HCC) based on serum biomarkers and imaging and sixty patients who were diagnosed with the hepatitis C virus (HCV) based on laboratory findings and imaging were included in this study. Thirty healthy individuals as a control group were also included in the study with matched age and sex. Demographic data and medical history were recorded for each patient.


**Inclusion criteria:**


Patients aged above 18 years.

Genders of both sexes were included.


**Exclusion criteria:**


Any cancer other than HCC.

Patients who were to receive chemotherapy or radiation.

### 2.3. Blood Samples

5 mL of blood was withdrawn from each participant using sterile labeled vacutainers. Samples were centrifuged at 3000× *g* for 10 min and sera were stored at −80 °C until RNA extraction.

### 2.4. miR-122 (NR_029667) and LncRNA HULC (NR_004855.2) Gene Expression Analysis

Total RNA was extracted from serum samples using miRNeasy mini kit and protocol for purification of total RNA, including miRNA (Qiagen, Hilden, Germany, Cat. No. 217004). Extracted RNA was subjected to RNA quantitation and purity assessment using NanoDrop^®^ (ND)-1000 spectrophotometer (NanoDrop technologies, Inc., Wilmington, DE, USA).

Reverse transcription: This step was carried out using miScript^®^ II RT kit and protocol for RNA reverse transcription into cDNA (Qiagen, Germany, Cat. No. 218161) as a part of miScript PCR system, which uses total RNA that contains non-coding RNA as the starting material for cDNA synthesis.


**PCR primers**


Target-specific primers assay for miRNA-122 and long non-coding RNA HULC were supplied by Qiagen, Germany. SNORD68 and GAPDH (as endogenous housekeeping genes) were used in this step. qRT-PCR was carried out using miScript SYBR^®^ Green PCR kit and protocol for miRNA and long non-coding RNA quantitative detection (Qiagen, Hilden, Germany, Cat. No. 218073). All samples were analyzed using the Rotor-Gene Q Real-Time PCR System (Qiagen, Valencia, CA, USA). The 2^−△△Ct^ method was conducted for the analysis and measurement of relative gene expression levels [7].

### 2.5. Research Ethics Statement

This study was approved by the Research Ethics Committee of the Faculty of Medicine, Helwan University, Egypt [Serial No. 23-2022]. The study was conducted according to the guidelines of the Declaration of Helsinki.

### 2.6. Statistical Analysis

Data analysis was performed using statistical package of social science (SPSS 17.0) on windows 8.1. Arithmetic means were calculated as central tendency measurement, while standard error was used as measure of dispersion for quantitative parametric data. Different tests were used for the quantitative parametric data, independent student *t*-test and one-way ANOVA test, one used for comparing two independent groups and the other for comparing more than two independent groups. To test significance, Bonferroni post-hoc was used. For the non-parametric data, Kruskal–Wallis and Mann–Whitney tests were used to compare more than two independent groups and to test the significance between groups. For measuring the correlation between groups, a bivariate Pearson correlation test, which was two-tailed to test the significance, was used. Sensitivity and specificity tests were generated to assess the new diagnostic tests with ROC Curve (receiver operating character). *p*-value < 0.05 was considered as a cutoff value for significance [8].

## 3. Results

### 3.1. Patients’ Demographics and Clinico-Pathological Data

Analysis of the demographic data of the studied groups showed that age was significantly higher in the HCC group than the HCV and control groups (*p* < 0.001, =0.001, respectively). There was also a significant difference in age between the HCV group and the control group (*p* = 0.029). There was no significant difference in gender among the studied groups (Table 1).

Plasma ALT and AST showed the highest levels in the HCC group, which was significantly different to the HCV group (*p* = 0.045, 0.001, respectively) and control group (*p* = 0.001). Furthermore, there was significant difference between the HCV and control group (*p* = 0.001). Serum bilirubin (total and indirect) was elevated significantly in the HCC group compared to the HCV group (*p* = 0.013, 0.008, respectively). CBC revealed anemia in the HCC group, manifested by a decreased Hb concentration, which was significantly different to the HCV and control groups (*p* = 0.021, 0.0001, respectively). The HCC group showed thrombocytopenia, with its platelet count being significantly lower than the HCV and control groups (*p* = 0.001). Moreover, it was significantly lower in the HCV group than in the control group (*p* = 0.002). Serum urea was significantly lower in the HCC group than in the HCV and control groups (*p* = 0.001, 0.05 respectively), but it was insignificantly different between the HCV and control groups. α-fetoprotein was significantly elevated in the HCC group compared to the HCV group with marked significant difference (*p* = 0.0001). There was no significant difference in serum ALP, serum albumin, or the total leukocytic count among the studied groups (Table 2).

Analysis of the clinical and pathological features of HCV patients revealed ascites in 36.6% of cases, with 10% having mild ascites, 13.3% having moderate, and 13.3% having marked ascites. A liver pathological examination was used to assess the grade of inflammation (activity) and the extent of fibrosis (stage) using the METAVIR scoring system, which categorized the patients into 80% A1, 10% A2, and 10% A3. Liver fibrosis stages among this group were 60% F1, 26.7% F2, and 13.3% F3 (Figure 1).

Regarding the HCC group, diabetes was present in 12 (40%) of the cases, and hypertension was present in 14 (46.7%) of the cases. Ascites was detected in 33.3% of the cases and was categorized as: mild in three (10%) cases, moderate in five (16.6%) cases, and marked in two (6.7%) cases (Figure 2).

### 3.2. Gene Expression Analysis of Serum miR-122 and LncRNA HULC and Their Diagnostic Potential in HCV

The enhanced gene expression of both miR-122 and long non-coding RNA HULC was noted in patients with HCV. There was no significant correlation between any of the studied biomarkers and the demographic or the laboratory findings in HCV patients. However, a positive correlation was detected between miR-122 and the serum level of AFP (Figure 3a). The level of gene expression of both biomarkers was found to be related to the grade of liver inflammation (activity). This manifested by the enhanced expression of miR-122 in A3 patients compared to A1 and A2 patients (*p* = 0.05, 0.04, respectively). HULC was also significantly higher in A3 patients than in A2 patients (*p* = 0.05) (Figure 3b). Regarding the relation between serum biomarkers and liver fibrosis stages in HCV patients, miR-122 was significantly lower in F3 patients than in F2 patients (*p* = 0.05), HULC was significantly higher in F3 patients than in F2 patients (*p* = 0.03), but there was no significant difference between patients with other fibrosis stages (Figure 3c). No significant difference was detected between serum biomarkers and the degree of ascites in HCV patients.

The diagnostic value of miR-122 and HULC was determined by ROC curve analysis. miR-122 was found to be a significant predictor of HCV (AUC: 0.997, 95%; CI: 0.989–1.00; *p* < 0.0001), at a cut-off value of 8.16 it has a sensitivity of 96.8%, specificity of 98.9%, and accuracy of 97.85%. HULC was also found to be a potential diagnostic biomarker for HCV (AUC: 0.847, 95%; CI: 0.684–1.00; *p* < 0.001); at a cut-off value of 16.25 it has a sensitivity of 84.2%, specificity of 99.1%, and accuracy of 91.65% (Figure 4a,c).

The relation between the serum biomarkers for HCV patients was determined by using a linear regression model (with miR-122 as the dependent variable): the results showed that there was a relation between miR-122 and HULC where *p*-value = 0.005, the unstandardized coefficient was −0.115, and the standardized coefficient was −0.206 (Figure 5a).

### 3.3. Gene Expression Analysis of Serum miR-122 and LncRNA HULC and Their Diagnostic Potential in HCC

The enhanced gene expression of both miR-122 and long non-coding RNA HULC was detected in the HCC group. The miR-122 level was higher in the HCV group compared to the HCC group with significant difference (*p* = 0.004). While HULC was significantly higher in the HCC group compared to the HCV group with significant difference (*p* = 0.018) (Figure 6), there was no significant correlation between any of the studied biomarkers and the demographic or the laboratory findings in HCC patients. Moreover, no significant correlation was found between the tested biomarkers and the clinical findings in HCC patients.

The ROC curve analysis revealed that miR-122 can be a significant predictor for HCC (AUC: 0.993, 95%; CI: 0.977–1.00, *p* = 0.001); at a cut-off value of 2.11 it has a sensitivity of 97.77%, specificity of 96.8%, and accuracy of 97.29%. HULC showed a diagnostic value for HCC (AUC: 0.571, 95%; CI: 0.355–0.788, *p* = 0.05); at a cut-off value of 29.8 it has a sensitivity of 57.1%, specificity of 100%, and accuracy of 78.55% (Figure 4b,d).

The relation between the serum biomarkers for HCC patients was determined with a linear regression model using miR-122 as the dependent variable. They were found to be negatively correlated in the HCV group with the unstandardized coefficient −0.115 (Figure 5a) and not correlated in the HCC group with the unstandardized coefficient 0.009 (Figure 5b).

## 4. Discussion

Hepatocellular carcinoma (HCC) represents a significant clinical problem, being the second leading cause of cancer deaths in the world. The risk of developing HCC is 17-fold higher in HCV-infected patients [9]. Persistent infection of the human liver with HCV can cause chronic hepatitis and cirrhosis, which are frequently followed by hepatocellular carcinoma (HCC) [10]. Chronic HCV infection is the most important risk factor in developing HCC in Egypt [11].

The high prevalence of HCC requires the continuous search for specific, accurate, and non-invasive biomarkers for early detection and follow up, leading to better prognostic outcomes. Researchers have done a lot of work in this field, which still requires significant efforts to validate everyday investigated biomarkers.

Hence, we have decided to search for reliable biomarkers with which to diagnose both HCC and HCV. This work will possibly complement previous studies that aimed at attaining better patient therapeutic outcomes.

Non-coding RNAs, including LncRNAs and microRNAs, play an important role in the translational regulation of gene expression, as well as alternative splicing [12]. MicroRNAs play a vital role in the sequence-specific negative regulation of the translation and stability of target mRNAs [13] and are considered the most well-studied epigenetic regulators in HCC [14]. HULC is considered to be the most overexpressed of LncRNAs in human HCC [15]. miR-122 is the most abundantly expressed miRNAs in the normal human liver. Its expression is reduced in HCC [16] and is considered a marker for poor prognosis [17].

In this work, we analyzed and compared the gene expression of LncRNA HULC and miR-122 in HCC patients (predisposed to HCV) and patients with HCV. The level of gene expression was correlated with clinical and laboratory findings. Moreover, ROC curves were established and analyzed to demonstrate the diagnostic and prognostic potentials of these biomarkers. The study included 120 patients divided into two groups; Group 1 included 60 patients diagnosed with HCC (50 males and 10 females) with a mean age of 60.47 ± 7.28 years, and Group 2 included 60 patients diagnosed with HCV (48 males and 12 females), with a mean age 38.20 ± 8.20 years. In addition, 30 healthy controls (19 males and 11 females) were included in this study with a mean age of 34.47 ± 3.70 years.

Analysis of the demographic data of the studied groups showed that age was significantly higher in the HCC group than in the HCV group. There was no significant difference in gender among the studied groups (Table 1).

Interpretation of the clinical data (Table 2) showed that plasma ALT and AST were significantly elevated in the HCC group and significantly different from the HCV group and control group. Moreover, there was significant difference between the HCV and control groups. Serum bilirubin (total and indirect) was significantly elevated in the HCC group compared to the HCV group. α-fetoprotein was significantly elevated in the HCC group compared to the HCV group with marked significant difference. There was no significant difference in serum ALP or serum albumin among the studied groups. Serum urea was significantly lower in the HCC group than in the HCV and control groups, but its difference was insignificant between the HCV and control groups.

Other studies showed higher levels of ALT, AST, total serum bilirubin [9,18], AFP [9,18,19,20], and lower serum albumin [9,18] among HCC patients, with statistically significant difference in the HCV group.

The HCC group presented a decreased Hb concentration, which was significantly different from the HCV group. Moreover, it presented thrombocytopenia, with a platelet count significantly lower than the HCV group (Table 2). These results were consistent with previous studies [9,19,21].

It is to be noted that a low Hb level is associated with mortality independently from the tumor stage, age, gender, and C-reactive protein levels. Anemia should be considered as a risk factor for mortality in HCC patients [22]. A low platelet count predicts liver fibrosis and cirrhosis, being lower with the further progression of cirrhosis; thus, lower platelet counts are found in HCC cases [19].

Regarding the associated pathological features, in the HCV group ascites was detected in 36.6% of cases, with 10% having mild ascites, 13.3% having moderate, and 13.3% having marked ascites. A liver pathological examination was used to assess the grade of inflammation (activity) and the extent of fibrosis (stage) using the METAVIR scoring system, which categorized the patients into 80% A1, 10% A2, and 10% A3. Liver fibrosis stages among this group were 60% F1, 26.7% F2, and 13.3% F3 (Figure 1).

As for the HCC group, diabetes was present in 40% of cases, and hypertension was present in 46.7% cases. Ascites was detected in 33.3% of cases and categorized as mild in 10% of cases, moderate in 16.6% of cases, and marked in 6.7% of cases (Figure 2).

Currently, it is known that miR-122 regulates various physiological and pathological processes within hepatic cells, such as lipid metabolism, response to drug or alcoholic hepatic injury, response to viral infection, and hepatic fibrosis formation [23]. It is suggested that miR-122 may function as a tumor suppressor during the process of hepatocarcinogenesis [24]. It was demonstrated that miR-122 targets the oncogenic SerpinB3 gene and thus can be of therapeutic benefit in HCC [25].

Interestingly, higher levels of circulating miR-122 have been observed in HCC patients versus those without HCC, suggesting serum miR-122 as a potential biomarker of HCC [26]. Controversy surrounding this theory might be due to the different sample sizes or races of study populations. However, a general problem with circulating microRNA assays is the lack of normalization of the results. There are currently no commonly accepted internal references for circulating microRNAs [27].

In our work, the enhanced gene expression of miR-122 was detected in both HCV and HCC groups, with miR-122 levels higher in the HCV group compared to the HCC group with significant difference (Figure 6). This was consistent with another study [28], which reported that levels of miR-122 in the serum of patients with chronic hepatitis were significantly higher than those in patients with HCC. A positive correlation was found between miR-122 and the serum level of AFP in the HCV group (Figure 3a), but no significant correlation was detected with other data (age, gender, ALT, AST, ALP, albumin, total bilirubin, indirect bilirubin, Hb, WBCs, platelets, urea, and creatinine). This was in agreement with a previously conducted study [27]. Moreover, a previous work [29] concluded that serum microRNA-122 levels markedly decreased while ALT significantly increased with increasing fibrosis stages during CHC infection.

On the other hand [30], highly observed significant correlations were found between serum miR-122 levels and alanine aminotransferase (ALT), aspartate aminotransferase (AST), γ-glutamyl transferase (GGT), and alkaline phosphatase (ALP) (parameters of liver cell damage). Moreover, there was a negative correlation between serum miR-122 levels and the international normalized ratio (INR), an indicator of liver function. In contrast, there were no significant relations found between miR-122 serum levels and the serum bilirubin, serum albumin, and total serum protein levels. There was a negative correlation between the serum levels of miR-122 and creatinine.

In the current study, in the HCC group, no significant correlation was detected between the expression of miR-122 and age, gender, ALT, AST, ALP, albumin, total bilirubin, indirect bilirubin, Hb, WBCs, platelets, AFP, urea, and creatinine.

HULC is a long non-coding RNA, known to be specifically expressed in hepatocytes, and which plays an important role in tumorigenesis [31]. It is involved in the pathogenesis of HCC with a highly upregulated expression in HCC compared to in controls [32,33]. HULC enhances liver carcinogenesis by stimulating CyclinD1 and inhibits P21 WAF1/CIP 1 via the autophagy-miR675-PKM2 pathway in human liver cancer stem cells [34].

In the present study, there was an enhanced gene expression of long non-coding RNA HULC in both HCV and HCC groups, and HULC was significantly higher in the HCC group compared to the HCV group with significant difference (Figure 6).

A previous study [9] reported highly upregulated HULC expression in an HCC group compared to HCV and control groups and a non-significant increase in the HCV group compared to the control group. Moreover, another study [33] found that HULC was aberrantly upregulated in HCC tissues and associated with the tumor node and metastases (TNM) stage, HCC recurrence, intra-hepatic metastases, and postoperative survival.

In our study, the levels of gene expression of the studied biomarkers (miR-122 and HULC) were correlated to the grade of liver inflammation in patients with HCV. This was manifested by the enhanced expression of miR-122 in A3 patients compared to A1 and A2 patients, but it was insignificantly different between A1 and A2 patients. Regarding its relation to liver fibrosis stages, miR-122 was significantly lower in F3 patients than in F2 patients, but there was no significant difference between F1 and F2 patients or between F1 and F3 patients (Figure 3).

In previous studies [30,35], the serum levels of miR-122 in patients with hepatic decompensation were significantly lower compared to patients with compensated liver disease.

The reduced serum miR-122 is most likely the result of a reduced release from hepatocytes. In cirrhotic patients who lost a large proportion of hepatocytes and thus have less functional hepatic capacity, the release of miR due to damage might be lower than in patients with higher amounts of healthy liver tissue [30]. Another possibility is that miR-122 serum levels are reduced due to higher volume distribution in patients with ascites. This indicates that in patients with liver cirrhosis, the miR-122 serum level might be a marker for hepatic functional capacity, whereas at earlier stages of liver disease, the serum miR-122 level is mainly an indicator of necro-inflammatory activity and cell death in the liver as release from damaged hepatocytes might be the major source of hepatocyte-derived miRs [27].

In the HCC group, miR-122 was not correlated with age, gender, or any of the laboratory findings. Moreover, it was not correlated with tumor size, stage, or metastasis.

It is worth mentioning that HULC was significantly higher in A3 patients than in A2 patients, but it was insignificantly different between A1 and A2 patients and between A1 and A3 patients. Regarding the relation to liver fibrosis stages, HULC was significantly higher in F3 patients than in F2 patients, but there was no significant difference between F1 and F2 patients or between F1 and F3 patients (Figure 3). No significant difference was detected between miR-122 or HULC and the degree of ascites in HCV. However, in HCC patients, HULC was significantly higher in mild ascites than in moderate ascites patients, but it was insignificantly different between mild and marked ascites patients and between moderate and marked ascites patients.

Regarding the correlation between HULC and demographic and laboratory data in HCV and HCC patients, there was no significant correlation between HULC and age, gender, ALT, AST, ALP, albumin, total bilirubin, indirect bilirubin, Hb, WBCs, platelets, AFP, urea, or creatinine.

A previously conducted study [9] reported that there was a significant negative correlation between the level of HULC and hemoglobin, PLT, and albumin levels, while a significant positive correlation with AST and ALT levels was found, confirming that HULC can be used as a marker for diagnosing HCC.

The diagnostic value of miR-122 and HULC was determined by ROC curve analysis. miR-122 was found to be a significant predictor for HCV at a cut-off value of 8.16 with a sensitivity of 96.8% and a specificity of 98.9%. It was also found to be of diagnostic and prognostic potential for HCC at a cut-off value of 2.11 with a sensitivity of 97.77% and a specificity of 96.8% (Figure 4a,b).

Our results were in agreement with another study [28] as ROC curve analyses revealed that serum miR-122 was a potential marker for discriminating HCC patients from healthy controls, with cut-off values of 0.70; sensitivity and specificity were 70.7% and 69.1%. Similarly, ROC curve analyses revealed that miR-122 was a useful marker for distinguishing patients with chronic hepatitis from healthy controls with cut-off values of 1.50 and with a sensitivity and specificity of 80.0% and 91.2%, respectively.

It had been emphasized that the measurement of circulating miR-122 may confer moderate diagnostic efficacy for HCC according to histopathological examination, particularly for the distinguishing of HCC patients from healthy controls or patients with HBV or HCV infections [36].

In the present study, HULC was also found to be a potential diagnostic biomarker for HCV at a cut-off value of 16.25, with a sensitivity of 84.2% and a specificity of 99.1%. As for its diagnostic value for HCC, HULC showed a diagnostic potential at a cut-off value of 29.8, with a sensitivity of 57.1% and a specificity of 100% (Figure 4c,d).

A previous study [37] suggested that HULC has adequate sensitivity and specificity to discriminate between HCC and control samples.

Moreover, in another study [9], an ROC curve was used to predict HCC using the fold change of HULC and showed that the probability of being true positive is 94.4% more than being false positive by repeating the test 100 times with a sensitivity of 94.4% and a specificity of 95% at a cut-off point of 1.25. Therefore, they concluded that HULC fold changes below this cut-off value are mostly predictive of HCC.

A linear regression model was constructed to determine the relation between the tested biomarkers, and they were found to be negatively correlated in the HCV group with the unstandardized coefficient −0.115 (Figure 5a) and not correlated in the HCC group with the unstandardized coefficient 0.009 (Figure 5b).

## 5. Conclusions

Our study demonstrated the expression of lncRNA HULC and miR-122 in HCC patients compared to HCV patients and a normal control.miR-122 was markedly expressed in the HCV group compared to the HCC group. It can be considered a non-invasive diagnostic biomarker for HCV with a sensitivity of 96.8% and specificity of 99%.miR-122 was significantly more highly expressed in the HCV group compared to the HCC group, indicating its prognostic value as a predictor of HCC in patients with chronic HCV.HULC was markedly expressed in the HCC group compared to the HCV group and control group. It can be considered a non-invasive diagnostic biomarker for HCC with a sensitivity of 57% and specificity of 100%.HULC was also expressed in the HCV group; hence, it can be considered a non-invasive diagnostic biomarker with a sensitivity of 84% and specificity of 99%.It is recommended to use other miRNAs to make an miRNAs signature through the development of panels containing many miRNA biomarkers and LncRNAs, or to add even conventional α-fetoprotein to these panels in the diagnosis and screening of HCV and HCC, which would help achieve greater accuracy.

## Figures and Tables

**Figure 1 genes-13-01669-f001:**
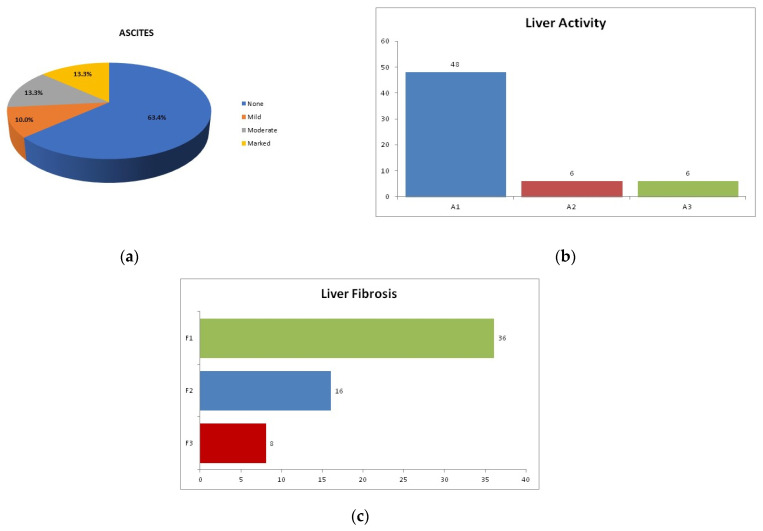
Clinical findings in HCV patients. (**a**) percentage of HCV patients with ascites, (**b**) liver activity, and (**c**) liver fibrosis stages.

**Figure 2 genes-13-01669-f002:**
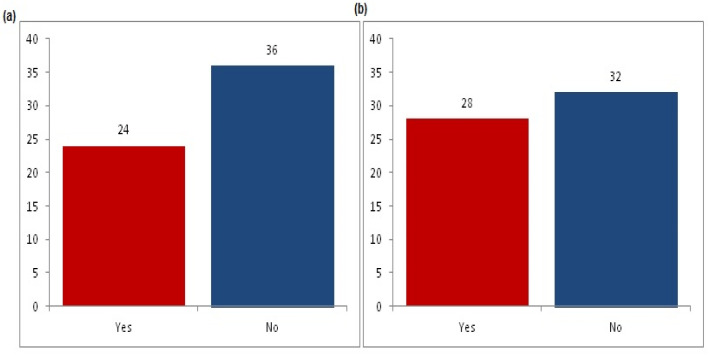
Clinical findings in HCC patients. (**a**) diabetes mellitus, (**b**) hypertension, and (**c**) percentage of HCC patients with ascites.

**Figure 3 genes-13-01669-f003:**
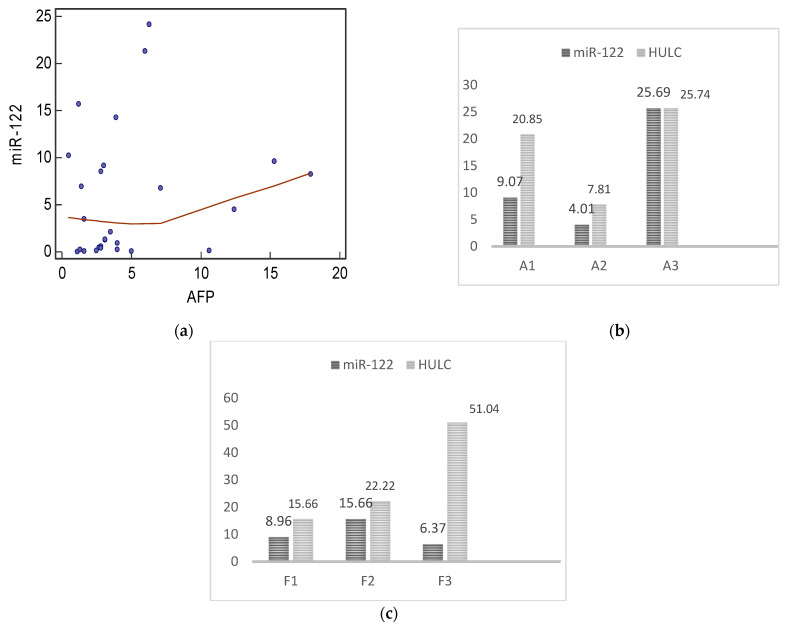
Correlation between gene expression of biomarkers and clinico-pathological findings in HCV patients. (**a**) correlation between miR-122 and AFP (r = 0.703, *p* = 0.001), r Pearson correlation, *p* value is statistically significant at *p* ≤ 0.05, (**b**) relation between serum biomarker expression levels and liver activity (A1, A2, and A3) among HCV patients, values are mean ± SE, miR-122 is significantly different between A1 and A3 (*p* = 0.05) and between A2 and A3 (*p* = 0.04), HULC is significantly different between A2 and A3 (*p* = 0.05), and (**c**) relation between serum biomarker expression levels and the extent of liver fibrosis (F1, F2, and F3) among HCV patients, values are mean ± SE, *p* = 0.05 between F2 and F3 in measuring miR-122 and *p* = 0.03 between F2 and F3 in measuring HULC.

**Figure 4 genes-13-01669-f004:**
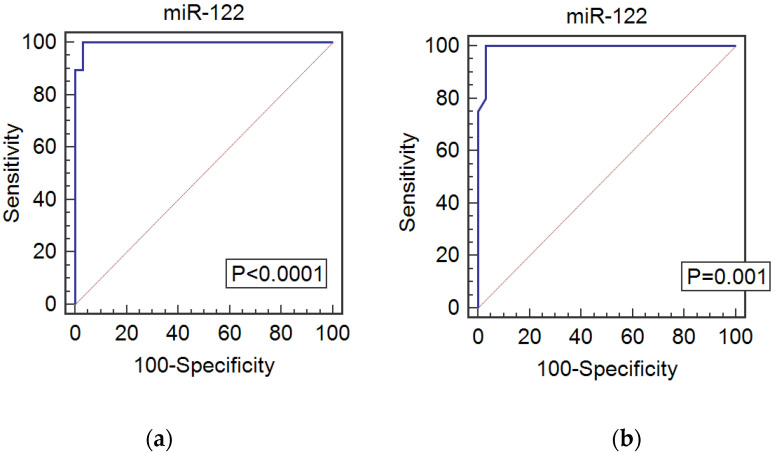
ROC curve analysis for miR-122 and HULC genes. (**a**) diagnostic performance of miR-122 for HCV, best cut-off value = 8.16 [sensitivity = 96.8%, specificity = 98.9%, and accuracy 97.85%], area under the curve (AUC) = 0.997, 95% confidence interval = 0.989–1.00, *p* < 0.0001, (**b**) diagnostic performance of miR-122 for HCC, best cut-off value = 2.11 [sensitivity = 97.77%, specificity = 96.8%, and accuracy 97.29%], area under the curve (AUC) = 0.993, 95% confidence interval = 0.977–1.00, *p* < 0.0001, (**c**) diagnostic performance of HULC for HCV, best cut-off value = 16.25 [sensitivity = 84.2%, specificity = 99.1%, and accuracy 91.65%], area under the curve (AUC) = 0.847, 95% confidence interval = 0.684–1.00, *p* < 0.001, and (**d**) diagnostic performance of HULC for HCC, best cut-off value = 29.8 [sensitivity = 57.1%, specificity = 100%, and accuracy 78.55%], area under the curve (AUC) = 0.571, 95% confidence interval = 0.355–0.788, *p* = 0.05.

**Figure 5 genes-13-01669-f005:**
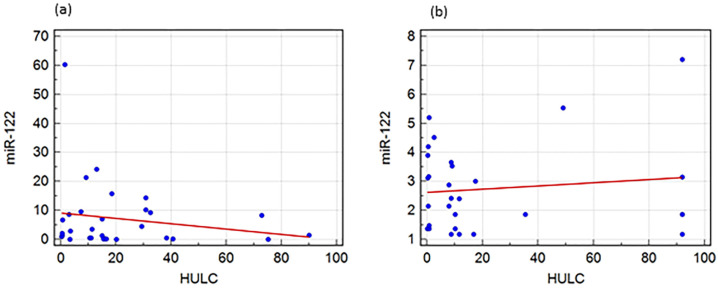
Logistic regression to determine the possibility of using the tested biomarkers in predicting HCV and HCC. (**a**) relationship between serum biomarkers for HCV group (unstandardized coefficient is −0.115, standardized coefficient is −0.206, *p*-value = 0.005) and (**b**) relationship between serum biomarkers for HCC group (unstandardized coefficient is 0.009, standardized coefficient is 0.194, *p*-value = 0.0001).

**Figure 6 genes-13-01669-f006:**
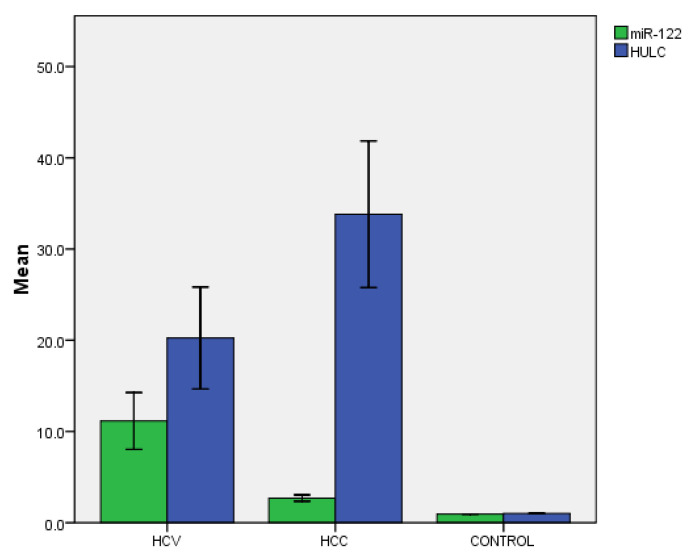
Mean levels of serum biomarkers among studied groups. Mean RQ value of miR-122 in HCV group = 11.16 ± 3.11 and in HCC = 2.69 ± 0.35 with *p* = 0.004. Mean RQ value of HULC in HCV group = 20.25 ± 5.58 and in HCC = 33.80 ± 8.03 with *p* = 0.018.

**Table 1 genes-13-01669-t001:** Demographic data of the studied groups.

Clinical Data	HCV (N = 60)	HCC (N = 60)	Control (N = 30)	*p*-Value
Age (years)	38.20 ± 8.20	60.47 ± 7.28	34.47 ± 3.70	**0.0001 a** **0.029 b** **0.001 c**
Gender				
Female	12 (20%)	10 (16.7%)	11 (36.7%)	
Male	48 (80%)	50 (88.3%)	19 (66.3%)	**0.15**

Age is shown as mean − SD, *p*-value (age) one-way ANOVA. Gender is presented by *n* (%) *p*-value (gender) chi-squared test. *p* values in bold are statistically significant (*p* ≤ 0.05). (a) between HCV and HCC, (b) between HCV and control, and (c) between HCC and control.

**Table 2 genes-13-01669-t002:** Laboratory findings in the studied groups.

Clinical Data	HCV (N = 60)	HCC (N = 60)	Control (N = 30)	*p*-Value
ALT (U/L)	59.6 ± 41.8	84.8 ± 52.5	30.2 ± 6.6	**0.045 a** **0.001 b** **0.000 c**
AST (U/L)	49.03 ± 36.1	114.6 ± 65.5	25.70 ± 5.8	**0.0001 a** **0.001 b** **0.000 c**
ALP (U/L)	151.2 ± 159.3	184.3 ± 162.6	-	0.43
Albumin (g/dL)	3.81 ± 0.73	3.5 ± 1.16	-	0.26
Total Bilirubin (mg/dL)	1.02 ± 0.66	1.635 ± 1.1147	-	**0.013**
Indirect Bilirubin (mg/dL)	0.33 ± 0.356	0.71 ± 0.65	-	**0.008**
Hb (g/dL)	13.30 ± 1.9	11.24 ± 2.5	13.32 ± 1.22	**0.021 a** 0.20 b **0.0001 c**
TLC x1000/µL	6.6 ± 2.30	5.5 ± 2.2	6.2 ± 1.41	0.077 a 0.39 b 0.10 c
Plts x1000/µL	206.37 ± 96.36	119.60 ± 52.25	296.6 ± 61.8	**0.001 a** **0.002 b** **0.001 c**
AFP (ng/mL)	5.71 ± 6.63	1449.08 ± 2911.0	-	0.0001
Urea (mg/dL)	25.89 ± 6.07	11.39 ± 3.38	29.16 ± 6.06	**0.001 a** 0.20 b **0.05 c**
GGT (U/L)	-	68.79 ± 60.9	-	**-**
Creatinine (mg/dL)	0.84 ± 0.17	1.08 ± 0.43	0.78 ± 0.13	**0.04 a** 0.1 b **0.01 c**

Values are shown as mean ± SD, *p*-value (age) one-way ANOVA. *p* values in bold are statistically significant. (*p* ≤ 0.05). (a) between HCV and HCC, (b) between HCV and control, and (c) between HCC and control. ALT = alanine transferase, AST = aspartate transferase, ALP = alkaline phosphatase, TLC = total leukocytic count, Plts = platelets, GGT = γ glutamyl transferase, and AFP = α-fetoprotein.

## Data Availability

Not applicable.

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
