# Peer review of "LncRNA HULC and miR-122 Expression Pattern in HCC-Related HCV Egyptian Patients"

_genes, 2022, doi:10.3390/genes13091669_

Round 1

Reviewer 1 Report

-       In line 32, comma is not required

-       In line 68, “The lncRNAs,…” should be rephrased for clarity

-       In line 76, “Scientist have…”

-       Are the HCC patient pool treatment naïve because any kind of  treatment can potentially alter the biomarker profile and confound the results? This needs to be addressed 

-       In line 117, Serial No. is missing

-       What happens to HULC lncRNA during general liver infection or inflammation, not necessarily HCC? This needs to be addressed.

Author Response

Point 1:   In line 32, comma is not required

Response 1: corrected (now it is line 30 after layout editing)

Point 2: In line 68, “The lncRNAs,…” should be rephrased for clarity

Response 2: The lncRNAs, miRNAs (21-24 bases) and pi RNAs (26-31 bases) function as regulators of gene expression on both the transcriptional and post-transcriptional level. Also, they play a role in DNA epigenetic modification.

 (now it is line 53)

Point 3:  In line 76, “Scientist have…”

Response 3: Done (line 60)

Point 4:    Are the HCC patient pool treatment naïve because any kind of treatment can potentially alter the biomarker profile and confound the results? This needs to be addressed 

Response 4: patients who received chemotherapy or irradiation were excluded from the study as mentioned in line 83.

Point 5:    In line 117, Serial No. is missing

Response 5: serial number was added  (line 106)

Point 6:     What happens to HULC lncRNA during general liver infection or inflammation, not necessarily HCC? This needs to be addressed.

Response 6: as mentioned in lines 169, 170&172, HULC expression is increased in HCV and the expression level is increased as inflammation increases.

Reviewer 2 Report

 Congratulations to the Authors. The study is very well designed but there is need to improve in the writing and results showcase.

1. There is different font size used in first two paragraphs.

2. Last paragraph should write before the last second paragraph to mention the rationale and hypothesis of the study. Do the patients had any other morbidities?

3. In methods, please explain the process of serum for RNA isolation before using kit. Please mention the primer sequences of miR-122 and lnHULC for qRT-PCR.

4. In many figures and text paragraphs, texts are not in the proper page size layout so they are getting missing in the prints. It needs to correct.

5. Line-198- words are repeated "but there was no".

6. uppercase and lower case need to rewrite in figure 3 legend. Statistics are missing in figure 3b-c. Writing format of miR-122/microRNA122 is not consistent throughout the manuscript. It needs to recheck and correct.

7. Importantly, is there any set of genes/ gene/ transcription factors that miR-122/lncHULC bind to 3'UTR and modulate their expression. It is very easy to predict by using available online software like targetscan, miRdeep. The expression of selected top targeted gene/transcription factor can be confirmed and compared in healthy control and best responder  HCC tissues of patients and can corelate these results to the expression of miR-122.

Author Response

Point 1:   There is different font size used in first two paragraphs.

Response 1: corrected

Point 2: Last paragraph should write before the last second paragraph to mention the rationale and hypothesis of the study. Do the patients had any other morbidities?

Response 2: last paragraph was placed before the last second paragraph.

In line 166, the HCV patients comorbidities is mentioned as ascites,…etc

In line 180, the morbidities in HCC group is mentioned.

Point 3:  In methods, please explain the process of serum for RNA isolation before using kit. Please mention the primer sequences of miR-122 and lnHULC for qRT-PCR.

Response 3: 5ml blood were withdrawn from each participant using sterile labeled vacutainers. Samples were centrifuged at 3000 xg for 10 min, sera were stored at -80°C till RNA extraction.

Primers for miR-122 &HULC were provided as ready mix primer assay by Qiagen. No sequence was provided.

Point 4:    In many figures and text paragraphs, texts are not in the proper page size layout so they are getting missing in the prints. It needs to correct.

Response 4: Done.

Point 5:    Line-198- words are repeated "but there was no".

Response 5: corrected (now it is in line 174 after page size layout edit)

Point 6:     uppercase and lower case need to rewrite in figure 3 legend. Statistics are missing in figure 3b-c. Writing format of miR-122/microRNA122 is not consistent throughout the manuscript. It needs to recheck and correct.

Response 6: upper case & lower case are corrected. Statistics are added. Format is corrected

Point 7:     Importantly, is there any set of genes/ gene/ transcription factors that miR-122/lncHULC bind to 3'UTR and modulate their expression. It is very easy to predict by using available online software like targetscan, miRdeep. The expression of selected top targeted gene/transcription factor can be confirmed and compared in healthy control and best responder  HCC tissues of patients and can corelate these results to the expression of miR-122.

Response 7: it was demonstrated in a previous study that miR-122 targets the oncogenic SerpinB3 gene and thus can be of therapeutic benefit in HCC. (line 308)

As determined by a previous study, HULC enhances liver carcinogenesis by stimulating CyclinD1 and inhibits P21 WAF1/CIP 1 via autophagy-miR675-PKM2 pathway in human liver cancer stem cells. (line 336).
